# Pleiotropic Effects of the Protease-Activated Receptor 1 (PAR1) Inhibitor, Vorapaxar, on Atherosclerosis and Vascular Inflammation

**DOI:** 10.3390/cells10123517

**Published:** 2021-12-13

**Authors:** Julian Friebel, Eileen Moritz, Marco Witkowski, Kai Jakobs, Elisabeth Strässler, Andrea Dörner, Daniel Steffens, Marianna Puccini, Stella Lammel, Rainer Glauben, Franziska Nowak, Nicolle Kränkel, Arash Haghikia, Verena Moos, Heinz-Peter Schutheiss, Stephan B. Felix, Ulf Landmesser, Bernhard H. Rauch, Ursula Rauch

**Affiliations:** 1Charité Center 11—Department of Cardiology, Charité—University Medicine, 12203 Berlin, Germany; julian.friebel@charite.de (J.F.); WITKOWM@ccf.org (M.W.); kai.jakobs@charite.de (K.J.); elisabeth.straessler@charite.de (E.S.); andrea.doerner@charite.de (A.D.); daniel.steffens@charite.de (D.S.); marianna.puccini@charite.de (M.P.); stella.lammel@charite.de (S.L.); nicolle.kraenkel@charite.de (N.K.); arash.haghikia@charite.de (A.H.); ulf.landmesser@charite.de (U.L.); 2Berlin Institute of Health, 10178 Berlin, Germany; 3DZHK (German Centre for Cardiovascular Research), Partner Site Berlin, 10785 Berlin, Germany; 4Center of Drug Absorption and Transport, Institute of Pharmacology, University Medicine Greifswald, 17489 Greifswald, Germany; eileen.moritz@med.uni-greifswald.de (E.M.); bernhard.rauch@uni-oldenburg.de (B.H.R.); 5DZHK (German Centre for Cardiovascular Research), Partner Site Greifswald, 17475 Greifswald, Germany; stephan.felix@med.uni-greifswald.de; 6Department of Cardiovascular & Metabolic Sciences, Lerner Research Institute, Cleveland Clinic, Cleveland, OH 44195, USA; 7Medical Department I, Gastroenterology, Infectious Diseases and Rheumatology, Charité—University Medicine, 12203 Berlin, Germany; rainer.glauben@charite.de (R.G.); franziska.nowak@bih-charite.de (F.N.); verena.moos@charite.de (V.M.); 8Institute for Cardiac Diagnostics and Therapy (IKDT), 12203 Berlin, Germany; info@ikdt.de; 9Department of Internal Medicine B, Cardiology, University Medicine Greifswald, 17489 Greifswald, Germany; 10Department of Human Medicine, Section of Pharmacology and Toxicology, Carl von Ossietzky Universität, 26129 Oldenburg, Germany

**Keywords:** thrombo-inflammation, thrombin, atherosclerosis, protease-activated receptors, PAR1, toll-like receptors, endothelial activation, vascular inflammation, vorapaxar

## Abstract

Background: Protease-activated receptor 1 (PAR1) and toll-like receptors (TLRs) are inflammatory mediators contributing to atherogenesis and atherothrombosis. Vorapaxar, which selectively antagonizes PAR1-signaling, is an approved, add-on antiplatelet therapy for secondary prevention. The non-hemostatic, platelet-independent, pleiotropic effects of vorapaxar have not yet been studied. Methods and Results: Cellular targets of PAR1 signaling in the vasculature were identified in three patient cohorts with atherosclerotic disease. Evaluation of plasma biomarkers (*n* = 190) and gene expression in endomyocardial biopsies (EMBs) (*n* = 12) revealed that PAR1 expression correlated with endothelial activation and vascular inflammation. PAR1 colocalized with TLR2/4 in human carotid plaques and was associated with TLR2/4 gene transcription in EMBs. In addition, vorapaxar reduced atherosclerotic lesion size in apolipoprotein E–knock out (ApoEko) mice. This reduction was associated with reduced expression of vascular adhesion molecules and TLR2/4 presence, both in isolated murine endothelial cells and the aorta. Thrombin-induced uptake of oxLDL was augmented by additional TLR2/4 stimulation and abrogated by vorapaxar. Plaque-infiltrating pro-inflammatory cells were reduced in vorapaxar-treated ApoEko mice. A shift toward M2 macrophages paralleled a decreased transcription of pro-inflammatory cytokines and chemokines. Conclusions: PAR1 inhibition with vorapaxar may be effective in reducing residual thrombo-inflammatory event risk in patients with atherosclerosis independent of its effect on platelets.

## 1. Introduction

Thrombo-inflammation in atherosclerosis has been described as a complex interplay between blood coagulation and inflammation that plays a critical role in cardiovascular diseases (CVDs). Furthermore, thrombo-inflammation is associated with residual cardiovascular risk [1,2,3,4,5]. Thus, current guidelines suggest that dual-pathway inhibition with low-dose rivaroxaban for long-term secondary prevention should be considered for patients with high ischemic risk and low risk of bleeding [6,7,8].

Protease-activated receptors (PARs) comprise a family of four G protein-coupled receptors (PAR1–PAR4) that are activated by serine proteases derived from the coagulation cascade, including factor (F) Xa and thrombin (FIIa), immune cells, and pathogens [9,10,11]. Protease-activated receptors are considered nonclassical “pattern-recognition receptors” [12]. Furthermore, PAR signaling is linked to innate immunity via cooperation with toll-like receptors (TLRs) [12]. Protease-activated receptor 1 is the thrombin receptor expressed on platelets, but it is also expressed in endothelial cells (ECs), vascular smooth muscle cells (VSMCs), and immune cells [13,14,15,16]. Therefore, PAR1 signaling is central in mediating thrombo-inflammation.

Early studies found high PAR1 expression in human atherosclerotic lesions, suggesting a contribution of PAR1 to the pathophysiology of atherogenesis [15]. Since then, numerous studies have supported the concept that PAR1 plays a pathological role in inflammatory and infectious disease [14,17]. Genetic deletion of both PAR1 and a cell-penetrating PAR1 pepducin, PZ-128, caused a significant decrease in total atherosclerotic burden in apolipoprotein E–knockout (ApoEko) mice [18,19]. Targeting PAR1 upstream via FXa/FIIa inhibition showed vasoprotective effects [2]. Therapeutics that directly and specifically target PAR1 have not been evaluated in this context.

Vorapaxar, which selectively antagonizes PAR1, was approved by both the Food and Drug Administration (FDA) and the European Medicines Agency (EMA) as an add-on therapy to standard care antiplatelet treatment regimens for secondary prevention in patients with a history of acute myocardial infarction or peripheral artery disease. The TRA 2P–TIMI 50 trial demonstrated a significant benefit of vorapaxar in reducing cardiovascular death and ischemic events. Although mainly attributed to vorapaxar’s effect on platelet aggregation, the expression of PAR1 on ECs, VSMCs, and immune cells suggests additional non-hemostatic effects [3].

Hence, we evaluated specific targets within the vasculature of patients with atherosclerotic disease associated with PAR1-related thrombo-inflammation. Pleiotropic effects of vorapaxar on the development and progression of atherosclerosis in vivo have not yet been investigated, which may prevent full exploitation of vorapaxar’s anti-atherogenic effects in the clinical setting. Therefore, we tested the hypothesis that vorapaxar attenuates de novo atherosclerosis and vascular inflammation in ApoEko mice. Because murine platelets, in contrast to human platelets, do not express PAR1 [20,21], we aimed to investigate potential pleiotropic, platelet-independent, PAR1-specific mechanisms in a murine model with a focus on ECs, foam-cell formation, inflammation, TLR2/4 interaction, and thrombogenicity, which can be addressed by a drug already in clinical use.

## 2. Materials and Methods

### 2.1. Patient Studies

Target identification of PAR1-mediated thrombo-inflammation was made in three separate patient cohorts.

In 190 patients with chronic CVD who were recruited from our outpatient department, we measured plasma biomarkers (using ELISA) indicative of endothelial activation—vascular cell adhesion protein 1 (VCAM1), intercellular adhesion molecule 1 (ICAM1), and endothelial-leukocyte adhesion molecule 1 E-selectin—and vascular inflammation (TNF-α, IL-6, and CRP). We correlated these with the percentage of FIIa-activated PAR1-positive (in which the antibody detects fragments of activated PAR1-cleaved-Ser^42^ protein) circulating peripheral blood mononuclear cells (PBMCs) using flow cytometry [22].

Data from patients with coronary atherosclerosis (*n* = 12) were retrospectively generated from a data bank of the collaborative research network SFB19. Patients underwent transvascular right-ventricular EMB for histological, immunohistological, and virological examination because of suspected cardiomyopathy. Patients were included if histological analysis of the biopsy sample showed no evidence of infiltrative or inflammatory myocardial disease.

Total cardiac mRNA was isolated from one biopsy, and the expression of *PAR1*, *VCAM-1*, *tissue factor* (*TF*), and *TLR2/4* was determined with qPCR. A detailed description of each patient’s characteristics is given in Table 1.

Carotid endarterectomy (CEA) specimens were collected from five unselected patients who were available for sampling with no adjustment for confounding.

The local ethics committee approved the study protocols, which were performed in accordance with the ethical principles in the Declaration of Helsinki. Each patient gave written informed consent before participation in the study.

### 2.2. Experimental Mouse Study

Twelve 8-week-old ApoEko male mice (Apoetm1Unc with a C57BL/6 genetic background) purchased from Charles River Laboratories were fed for four months with a Western-type diet (WD)—ssniff ± vorapaxar (control group/vorapaxar group), Selleckchem (no. E15721-34; 21% crude fat, 2 mg/kg cholesterol ± vorapaxar 10 mg/kg WD). A Western-type diet and WD with vorapaxar was replaced every week. Vorapaxar had no influence on food intake, weight gain, or circulating cholesterol levels (Appendix A), suggesting that the observed effects of vorapaxar were not related to changes in lipid metabolism per se.

To test stability and functionality of vorapaxar, a short-term (7 days on diet) preliminary experiment was performed. Murine platelets do not express PAR1. However, by stimulating human platelets (PRP) in vitro with isolated plasma (PPP) from mice that received vorapaxar for 7 days, we confirmed the presence of vorapaxar and its PAR1-inhibiting capacity in biologically relevant concentrations after oral uptake in ApoEko mice (Appendix A).

The average uptake of vorapaxar was approximately 1 mg/kg/d in mice, which reflects human exposure at the recommended dosage of 2.5 mg once a day. Animals were housed in standard conditions, according to the international guidelines of Directive 2010/63/EU of the European Parliament, and studies were approved by the local animal authority (Berlin, Germany; No. G0169/16).

### 2.3. Materials

A detailed description of the chemicals used is given in the Appendix A.

### 2.4. Histology, Immunostaining, and Confocal Microscopy

Sequential 10 µm aortic root sections were cut from the point of appearance of the aortic valve leaflets with a Leica CM3050S cryostat machine (Leica Biosystems, Wetzlar, Germany). Hematoxylin and eosin (HE) staining was used to quantify atherosclerotic lesion size in the aortic sinus coronaries (% lumen reduction by plaque area). Immunostaining was performed with specific antibodies (Appendix A). Fluorescence was detected using an LSM 510 Meta laser scanning microscope (Zeiss, Jena, Germany), and images were processed using the ZEN 2.3 SP1 software (Carl Zeiss, Jena, Germany).

Aortas were excised, cleaned from fat tissue, and stained with Oil red O. Oil Red O solution was used to visualize the extent of plaques in the aortic arch and lipid retention in the aortic sinus [23]. Image quantification was performed using ImageJ 1.52 h software (NIH).

### 2.5. qPCR

For quantitative PCR (qPCR), total mRNA was isolated with peqGOLD Trifast. The expression of indicated markers was analyzed with a FAM-tagged TaqMan^®^ gene expression assay. Relative gene expression was determined using the comparative C(t) (ΔΔCt) method with 18S ribosomal RNA as the endogenous control. Data are normalized to their respective control group.

### 2.6. Isolation of Endothelial Cells

Markers of endothelial activation and vascular inflammation (*VCAM1*, *E-selectin*, *TF*, *TLR2*, and *TLR4*) were assessed using real-time PCR in isolated murine ECs. First, the lungs were removed and incubated with collagenase with gentle agitation for 1 h in a 37 °C water bath. The resulting tissue/cell suspension was filtered through a 70 μm strainer, washed with phosphate-buffered saline, and centrifuged. The washed supernatant was incubated with prepared Dynabeads™ coupled with an anti-platelet EC adhesion molecule (PECAM-1/CD31) antibody for magnetic bead sorting.

### 2.7. Human Monocytic Cells (THP-1) Cells

Human monocytic cells (THP-1) were grown in RPMI 1640 medium + 10% FBS + 1% penicillin/streptomycin. THP-1 cells were exposed to 100 µg/mL–oxidized low-density lipoprotein (oxLDL) as the control in the presence of PAR1 agonist, thrombin (10 U/mL); TLR2 agonist, Pam3CSK4 (100 ng/mL); TLR-4 agonist, LPS (100 ng/mL); PAR1 inhibitor, vorapaxar (1 µM); and combined TLR2/4 inhibitor, candesartan (1 µM). Cells were immobilized onto glass microscope slides via cytospin, and lipoprotein uptake was measured by Picro-Sirius Red Stain (percent stained area/total cell area from 100 cells). Expression of *TLR2*, *TLR4*, *CD36*, *scavenger receptor type 1* (*SR-A1*), *lectin-type oxLDL receptor 1* (*LOX-1*), *ATP-binding cassette transporter* (*ABCA1*), and *ATP-binding cassette sub-family G member 1* (*ABCG1*) was assessed using real-time PCR.

### 2.8. ELISA

VCAM1, ICAM1, E-selectin, TNF-α, IL-6, CRP, P-selectin, platelet factor 4 (PF4)/C-X-C motif ligand 4 (CXCL4), thrombin/antithrombin complex (TAT), and free cholesterol ELISA were performed according to the manufacturer’s instructions.

### 2.9. Stimulation of Platelet-Rich Plasma and Platelet-Poor Plasma with Aortic Plaque Material

Pooled platelet-rich plasma (PRP) and platelet-poor plasma (PPP) were prepared as described by Vogt et al. [24] from mouse-citrated blood of six age-matched C57BL/6 (B6) wild-type (wt) mice. Atherosclerotic aortic tissues were freeze-dried and pulverized, and the tissue powders were subsequently dissolved [25]. The platelet-rich plasma and platelet-poor plasma were stimulated with 10 mg of homogenized aortic plaque material from either the control group or the vorapaxar-treated group, respectively.

### 2.10. Stimulation of Human PRP with PPP from ApoEko Mice

Platelet-poor plasma was isolated from ApoEko mice after one week on a WD ± vorapaxar. Pooled human PRP was stimulated with PPP from either control or vorapaxar-treated mice ± PAR1 agonistic peptide (AP) (TFLLR-NH2, 15 µM). P-selectin expression on platelets was measured via flow cytometry.

### 2.11. Statistical Analysis

After performing normality testing, single comparisons were assessed using a two-tailed unadjusted unpaired Student’s *t*-test. Differences among groups were analyzed with ANOVA followed by the Bonferroni-adjusted *t*-test. For correlation analysis, the Pearson coefficient was used. All analyses were performed using GraphPad Prism version 9.3.0 software. Results are expressed as single values ± SD. The overall α-level was 0.05.

## 3. Results

### 3.1. Target Identification of PAR1-Mediated Thrombo-Inflammation in Patients with Atherosclerotic Disease

#### 3.1.1. PAR1 Activation Corresponds to Endothelial Activation and Vascular Inflammation

The direct cellular downstream targets of the TF/FXa/FIIa/PAR1 axis, which go beyond hemostatic effects, have been identified only in preclinical studies. Therefore, we identified targets of PAR1-mediated thrombo-inflammation in patients with atherosclerotic disease. FIIa-activated PAR1 (the antibody detects PAR1 cleaved by FIIa)-positive PBMCs were used as an indicator for augmented signaling through the TF/FXa/FIIa-PAR1 axis [22]. Thrombin-cleaved PAR1 was associated with increased plasma biomarkers such as VCAM1, ICAM-1, and E-selectin in 190 high-risk patients (CRP of mg/L 7.4 ± 6.7 mean ± SD) with chronic CVD due to atherosclerosis. These markers are prognostically relevant and indicative of endothelial activation (Figure 1A). In addition, FIIa-activated PAR1-positive PBMC were correlated with pro-inflammatory markers such as TNF-α, IL-6, and CRP, showing vascular inflammation (Figure 1B). In line with these observations, PAR1 expression in EMBs from 12 patients with coronary atherosclerosis positively correlated with cardiac inflammatory cell infiltration (Mac-1) and the transcription of *VCAM-1* and *TF* (Figure 2A). In carotid atherosclerotic plaques, PAR1 is expressed within the endothelial layer and the immune cell-infiltrating area (Figure 3A).

#### 3.1.2. PAR1 Colocalizes with TLR2 and TLR4 in Human Atherosclerotic Disease

Activation of TLRs triggers endothelial dysfunction, contributes to early-stage atherosclerotic plaque formation, and mediates acute vascular adverse events [26,27]. Because PAR signaling is linked to innate immunity via cooperation with TLRs [12], we sought to evaluate their connection in atherosclerosis. Myocardial PAR1 expression strongly correlated with TLR2 and TLR4 in patients with coronary atherosclerosis (Figure 2B). Furthermore, immunofluorescence experiments revealed colocalization (% area of colocalization: mean ± SD 77.83 ± 19.47, *n* = 5) of PAR1 with TLR2 and TLR4 in carotid atherosclerotic lesions from patients who underwent carotid endarterectomy (CEA) (Figure 3B), pointing to a putative interaction of PAR1 with these innate immune receptors.

### 3.2. Evaluation of Pleiotropic, Vasoprotective Effects of the PAR1 Inhibitor Vorapaxar

Thus far, PAR1-mediated endothelial activation and vascular inflammation have been identified as potential therapeutic targets in patients with atherosclerotic disease. Additionally, vorapaxar thus far is the only specific and direct PAR1 inhibitor already in clinical use. The immediate next question for consideration: Is vorapaxar atheroprotective by reducing thrombo-inflammation?

#### 3.2.1. Vorapaxar Reduces de Novo Atherosclerosis in ApoEko Mice Fed an Atherogenic Diet

An established mouse model (mice have no PAR1 on platelets) of de novo atherosclerosis was used to study potential pleiotropic, platelet-independent effects mediated by PAR1. The average uptake of 10 mg/kg WD vorapaxar in mice reflects the human exposure at the recommended dosage of 2.5 mg once a day, which is indicated for secondary prevention. ApoEko mice developed lesions within the aortic arch after following an atherogenic diet (high fat and high sucrose = WD) for 16 weeks. (Figure 4A). Computer-assisted quantitative histomorphometric analysis revealed a 10% (aortic arch) to 30% reduction (aortic sinus) in the extent of atherosclerosis in mice that received vorapaxar orally (Figure 4A,B). Furthermore, inhibition of PAR1 resulted in decreased coronary artery stenosis (Appendix A).

#### 3.2.2. PAR1 Inhibition with Vorapaxar Attenuates Vascular Inflammation

Vascular inflammation is an important aspect during the pathogenesis of atherosclerosis and contributes to acute atherothrombotic complications [28]. Therefore, we examined the effect of vorapaxar on endothelial activation (Figure 5). Direct PAR1 inhibition via provision of vorapaxar reduced the expression of VCAM-1 and E-selectin (Figure 5A,B) in aortic specimens from ApoEko mice (relative mean fluorescence intensity of VCAM-1 in aortic plaques ± SD control vs. vorapaxar; *n* = 6: 1.0 ± 0.2 vs. 0.14 ± 0.08, *p* < 0.0001). Tissue factor contributes critically to in vivo atherothrombosis initiation and a procoagulant state [29,30,31,32,33,34,35,36]. In our model, treatment with vorapaxar also affected the aortic expression of *TF* (Figure 5A). These alterations in the expression of vascular inflammation markers in aortic tissue could be reproduced in isolated ECs from controls and vorapaxar-treated mice (Figure 5C). Furthermore, the endothelial expression of both inflammation-associated TLR2 and TLR4 was found to be reduced by vorapaxar in isolated ECs (Figure 5D) and aortic specimens (Figure 5E,F).

#### 3.2.3. PAR1 Inhibition with Vorapaxar Impairs Foam Cell Formation

Vascular inflammation mediates endothelial LDL translocation. Lipid uptake by resident or immigrated cells within the plaque, with subsequent foam cell formation and lipid accumulation, expedites the progression of atherosclerosis and the local inflammatory immune response. Vorapaxar treatment reduced lipid accumulation within aortic plaques of ApoEko mice (Appendix A). Therefore, we hypothesized that the thrombin receptor PAR1 is, together with TLR2/4, involved in intraplaque lipid metabolism. To address this hypothesis, the effect of thrombin on the oxLDL uptake in a human monocytic cell line (THP-1) was analyzed. Activation of PAR1 with thrombin increased monocytic lipid accumulation by 56%, an effect that was abrogated by vorapaxar (Appendix A). Treatment with TLR2 and TLR4 agonists enhanced thrombin-induced oxLDL retention (Appendix A). This may be related to a thrombin-mediated increase in TLR2 and TLR4 transcription and, therefore, enhanced TLR signaling capacity (Appendix A). Dual-target inhibition (the PAR1-inhibitor vorapaxar combined with the TLR2/4-inhibitor candesartan) synergistically reversed oxLDL accumulation (Appendix A). Immunofluorescence staining also revealed an increased expression and colocalization of PAR1 with TLR2 and TLR4 at the cell membrane following thrombin stimulation (Appendix A). This effect on innate immune receptor expression by thrombin was prevented by vorapaxar (Appendix A). In THP-1 cells, furthermore, thrombin upregulated the transcription of scavenger receptors *CD36*, *SR-A1*, and *LOX-1*, which are known to facilitate the uptake of lipoproteins (Appendix A). The transcription of *ABCA1* but not *ABCG1*—which both mediate cellular cholesterol efflux—in turn decreased upon PAR1 stimulation, indicating an increased lipid-retention capacity (Appendix A). Treatment of the cells with vorapaxar reversed the thrombin-induced transcriptional changes of receptors involved in cellular lipid accumulation (Appendix A).

#### 3.2.4. Vorapaxar Treatment Is Associated with an Anti-Inflammatory Cell and Cytokine Profile within Atherosclerotic Plaques

The results described indicate that vorapaxar has direct effects on the arterial wall phenotype and intraplaque lipid metabolism. Moreover, the immunological micromilieu and the balance between pro- and anti-inflammatory cells and cytokines are crucial for further plaque development and a continued adverse phenotype. The inhibition of PAR1 with vorapaxar led to a reduction in CD68-positive cells (pan- or M1 pro-inflammatory macrophage marker) that infiltrated the aortic plaque by 14% (Figure 6A,B). Atherosclerotic plaques were characterized by a predominance of inflammatory macrophages, as indicated by an increased ratio of CD80^+^/CD206^+^-infiltrating cells (M1 pro-inflammatory/M2 anti-inflammatory macrophages) (Figure 6A). This predominance was reduced by directly targeting PAR1 (Figure 6A). Likewise, the aortic expression of *CCL2*, the chemokine that is responsible for the recruitment of classical (inflammatory) monocytes into the plaque, was downregulated under vorapaxar treatment (Figure 6C). Vorapaxar also reduced the aortic gene expression of inflammation-associated chemokines (TNF-α, *IL-1β*) as well as the innate immune receptors *TLR2* and *TLR4* (Figure 6C). Furthermore, the presence of CD3^+^ T cells, along with inflammatory cytokines released by activated T cells (*IL-2*, *IL-12B*, *IFN-γ*), was found to be decreased by treatment with the PAR1 inhibitor (Figure 6D). A reduction in the necrotic core size and apoptotic plaque area, as observed in vorapaxar-treated mice, underlines the shift toward an anti-inflammatory immunological micromilieu (Figure 6E,F).

#### 3.2.5. PAR1 Inhibition with Vorapaxar Reduced Thrombogenicity in Mice

Thus far, we have described an atheroprotective, anti-inflammatory effect of vorapaxar in ApoEko mice on an atherogenic diet. Intraplaque inflammation initiates and maintains primary and secondary hemostasis, thereby provoking acute adverse vascular events. Because murine platelets do not express PAR1, we were able to study secondary (pleiotropic) effects on coagulation that are independent of vorapaxar’s capacity to affect platelet activation primarily. In addition, circulating plasma markers of platelet activation (P-selectin, PF4/CXCL4) and thrombin activity (assessed via TAT) were diminished under the treatment with vorapaxar (Figure 7A). Next, pooled PRP (to study platelet activation) and pooled PPP (to study thrombin generation), isolated from C57BL/6 wt mice, were stimulated with homogenized aortic plaque material of either control or vorapaxar-treated mice, respectively. The net thrombogenic potential of plaques from vorapaxar-treated mice was reduced, as indicated by a decreased release of platelet activation markers (P-selectin, PF4/CXCL4) in PRP and reduced thrombin generation (TAT) in PPP (Figure 7B). Therefore, we conclude that the observed vasoprotective effects by vorapaxar subsequently reduce thrombogenicity.

## 4. Discussion

### 4.1. Target Identification of PAR1-Mediated Thrombo-Inflammation in Patients with Atherosclerotic Disease

The FXa/FIIa/PAR1 inhibitors have been shown to therapeutically address residual thrombotic risk in patients with chronic CVD [1,2]. Pleiotropic actions of anticoagulants to provide additional vasoprotection have been postulated. Nevertheless, the direct cellular downstream targets of the TF/FXa/FIIa/PAR1 axis, which go beyond hemostatic effects, have been identified only in preclinical studies [2,3]. Therefore, by studying plasma biomarkers, EMBs, and carotid plaques, we highlighted PAR1-related endothelial activation and vascular inflammation as specific therapeutic targets in patients with atherosclerotic disease.

The FDA- and EMA-approved PAR1 inhibitor, vorapaxar, is a safe and selective PAR1 inhibitor available for clinical use, but its effect on atherosclerosis in vivo is unknown. This study investigated the functional effects of vorapaxar on the atherosclerosis phenotype, the vascular immune cell micromilieu, and the coagulation system in vivo. We found that providing vorapaxar to ApoEko animals improved plaque burden and vascular inflammation and promoted a vasculoprotective immune cell phenotype. Furthermore, PAR1 signaling was linked to innate immunity via cooperation with TLR2/4. Our studies provide novel evidence that vorapaxar has pleiotropic effects on the vasculature beyond inhibition of platelet function. Stabilization of plaque and reduced plaque thrombogenicity through PAR1 inhibition in our study are also likely to explain atherothrombotic risk reduction exhibited by drugs that inhibit upstream PAR signaling (e.g., FXa inhibitors).

### 4.2. The PAR1 Inhibitor Vorapaxar Attenuates de Novo Atherosclerosis

A strong expression of PAR1 has been confirmed in human and mouse atherosclerotic lesions (ECs, VSMCs, immune cells) [15,16,37]. Interestingly, the activity of PAR1 upstream coagulation proteases, FXa/FIIa, was enhanced in early-stage atherosclerotic plaques compared with advanced-stage plaques [25]. Genetic deletion of PAR1 in ApoEko mice has been shown to reduce atherosclerotic plaque progression [18]. Notably, a cell-penetrating PAR1 pepducin, PZ-128, caused a significant decrease in total atherosclerotic burden in ApoEko mice [19]. By translating these results into the atherosclerotic ApoEko mouse model, we observed that the clinically approved PAR1 inhibitor, vorapaxar, reduced atherosclerotic lesions in the aorta of ApoEko mice after four months on a WD.

### 4.3. PAR1 Inhibition with Vorapaxar Attenuates Vascular Inflammation

Endothelial activation and vascular inflammation are critical steps initiating atherosclerotic plaque development and have been linked to PAR1-dependent signaling [38,39]. Upregulation of adhesion molecules in ECs mediates the recruitment of immune cells that later are responsible for lipid retention and adverse thrombotic events [40]. Thus, PAR1 has been shown to be expressed in activated ECs of atherosclerotic plaque, but not in non-diseased cardiac arteries [15,16,41]. Previous studies demonstrated that vorapaxar prevented endothelial barrier disruption in cultured ECs [42,43,44,45]. Here, we report that vorapaxar reduced the expression of inflammation-associated adhesion molecules and TF, both in isolated ECs and the aorta in a murine model of atherosclerosis.

ApoEko mice, compared to low-density lipoprotein receptor-deficient (LDLRko) mice, have more pronounced endothelial dysfunction and endothelial disruption under the metabolic challenge of an atherogenic diet due to reduced protective activated protein-C-PAR1 signaling. This might explain why PAR1 deficiency had no anti-atherosclerotic effects in the LDLRko mouse model [37,46,47,48,49,50].

Activation of the endothelial innate immune pathways of TLR2 and TLR4 is known to modulate pathways involved in endothelial permeability and coagulation that are associated with the initiation and progress of atherosclerosis, as well as with acute atherothrombotic adverse events [26,27,51,52]. Transactivation of TLRs by PARs has been described to play an essential role during the innate immune response of the heart [53,54,55,56]. Besides direct transactivation, PAR1 might regulate the expression of endothelial TLRs. Here, we report a robust correlation of cardiac PAR1 and TLR2/4 expression in patients with coronary atherosclerosis. We also describe a colocalization of PAR1 with TLR2 and TLR4 in human atherosclerotic plaques. Furthermore, vorapaxar treatment in our study was associated with a reduced expression of TLR2 and TLR4 in isolated ECs and in the endothelial layer of the aorta from ApoEko mice.

Therefore, targeting PAR1 with vorapaxar might also affect endothelial inflammatory signaling that is mediated by antigen activation of pattern-recognition receptors during atherogenesis in ApoEko mice.

### 4.4. PAR1 Inhibition with Vorapaxar Impairs Foam Cell Formation

Vascular inflammation mediates endothelial LDL translocation. Lipid uptake by resident or immigrated cells within the plaque, with subsequent foam cell formation and lipid accumulation, expedites the progression of atherosclerosis and the local inflammatory immune response [40]. This process involves PAR1 [18,57,58,59]. In our model of diet-induced atherogenesis, vorapaxar reduced lipid retention in aortic plaques.

Previous studies suggest that TLR2 and TLR4 are responsible for specific atherogenic processes, such as foam cell formation and lipid accumulation [26,51]. Here, we show that thrombin increased *TLR2* and *TLR4* expression in THP-1 cells, thereby presumably potentiating their ability to accumulate lipids.

### 4.5. Vorapaxar Treatment Is Associated with an Anti-Inflammatory Cell and Cytokine Profile within Atherosclerotic Plaques

Intraplaque inflammation—which follows endothelial activation and lipid accumulation, which later mediates acute atherothrombotic events—is orchestrated by various cell types [40]. PAR1 has been found to be expressed by macrophages and T-lymphocytes in atherosclerotic plaques [15,16]. We found a reduction in total infiltrating pro-inflammatory macrophages, as well as a reduction of CD3^+^ T cells. This is paralleled by a decreased transcription of associated pro-inflammatory cytokines and chemokines. This suggests a net anti-inflammatory milieu in plaques of ApoEko mice that were treated with vorapaxar. There are two overlapping explanations for this observation. This could be a secondary effect to the observed attenuated vascular inflammation and foam cell formation, as outlined above. Second, vorapaxar directly affects inflammatory activation of macrophages and T cells [60,61,62].

Apoptosis of macrophages, foam cells, and other plaque-residing immune cells can be observed during plaque progression and transformation [40]. However, efficient clearance of these dead cells is essential for preventing secondary necrosis. Here, we observed a shift towards CD206^+^ cells. This macrophage subtype has been described as possessing high phagocytic activity, thereby presumably protecting against large necrotic core zones as observed in the vorapaxar-treated group. Furthermore, this subpopulation of macrophages has been shown to accumulate less oxidized and native LDL and possess a low susceptibility to becoming foam cells [63].

### 4.6. Clinical Implications

In TRACER and TRA 2P-TIMI 50, vorapaxar proved to efficiently reduce thrombotic cardiovascular events in patients with a history of acute myocardial infarction or peripheral artery disease in addition to single or dual antiplatelet therapy [64]. Therefore, it has been hypothesized that the observed residual ischemic risk is attributed to an upregulated status of thrombin-mediated platelet activation, which makes these patients more susceptible to the antithrombotic effects of vorapaxar or the FXa-inhibitor rivaroxaban [65].

Both animal studies and clinical trials have suggested pleiotropic non-hemostatic effects of PAR-affecting therapies on atherosclerosis or atherothrombosis. Pharmacological strategies targeting residual risk reduction in patients with atherosclerosis include a decrease in FXa/FIIa-mediated thrombo-inflammation [38,66,67]. In patients with coronary artery disease and peripheral arterial disease, the FXa-inhibitor rivaroxaban reduced atherothrombotic events as shown in the ATLAS ACS 2-TIMI 51 and COMPASS trials [64].

Because murine platelets lack PAR1 expression, our observation that circulating markers of platelet activation and thrombin generation are diminished under vorapaxar might be the consequence of pleiotropic effects on the endothelial layer. The inhibition of PAR1 with vorapaxar, reduced endothelial activation and inflammation during experimental human endotoxemia (bolus infusion of 2 ng/kg LPS) [68]. This is in accordance with the observation that PAR1 is present only at sites with activated endothelium—as is the case in (pre)atherosclerotic lesions, for instance—but not in non-diseased arteries [15,16,41].

Disruption of advanced atherosclerotic plaques, achieved by rupture or superficial erosion, does not automatically result in an acute thrombotic event that becomes clinically evident [69]. Therefore, a second hit has been postulated to be a prerequisite for a subocclusive or occlusive, clinically overt, superimposed thrombus formation. This can be caused by excessive plaque activity, blood thrombogenicity, and hypercoagulability [69,70,71,72]. An augmented inflammatory signaling via TLR2 and TLR4 can trigger such a second hit, thereby mediating an acute atherothrombotic event [26,73]. Recent studies have shown increased gut permeability with subsequent low-grade endotoxemia, enhanced artery thrombosis and platelet activation in a TLR4-dependent manner in patients with ST-elevation myocardial infarction (STEMI) [27,74,75].

The factors causative for the disruption of the plaque (first hit) and the factors associated with an insufficient healing process (second hit) might be potential targets of vorapaxar.

Interestingly, we found that homogenized plaque material from mice treated with vorapaxar exhibited a reduced capacity to activate platelets and generate thrombin. The observed change in plaque architecture and composition in our study might be associated with the net reduction of the thrombogenic potential of plaque material from mice treated with vorapaxar. Therefore, the authors speculate that vorapaxar may decrease the likelihood of an acute arterial event (caused by plaque rupture or erosion), and the subsequent hemostatic response will lead to a smaller thrombus and less-severe ischemic injury. There are, however, several important differences in plaque histology between humans and murine models that may limit the conclusions of this study.

## 5. Conclusions

Taken together, our findings contribute to the understanding of the atheroprotective effects of FXa/FIIa/PAR1-inhibitors observed in large clinical trials. In our study, vorapaxar exhibits pleiotropic effects on the vasculature beyond direct inhibition of platelet function. Therefore, PAR1-mediated thrombo-inflammation is a potential target to treat atherosclerosis and prevent adverse atherothrombotic events in patients with a high residual ischemic risk. The results of our study provide the rationale for prospective trials that consider individualized approaches for preventing acute adverse cardiovascular events.

## Figures and Tables

**Figure 1 cells-10-03517-f001:**
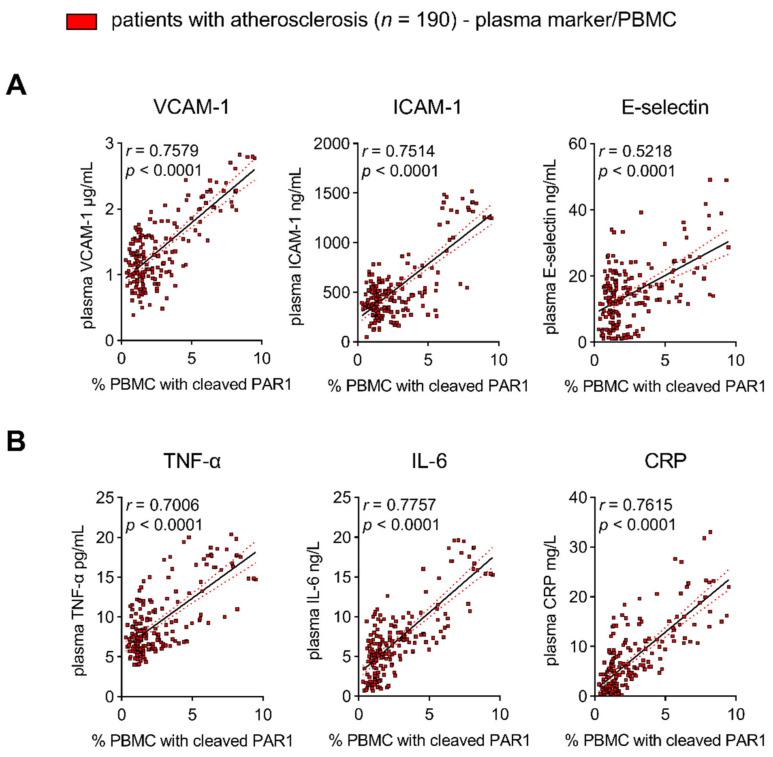
PAR1 activation corresponds to endothelial activation and vascular inflammation in patients with atherosclerotic disease. Active (thrombin-cleaved) PAR1 was detected via flow cytometry on circulating PBMCs. Biomarkers indicative of (**A**) endothelial activation (VCAM-1, ICAM-1, E-selectin) and (**B**) vascular inflammation (TNF-α, IL-6, CRP) were measured with ELISA. Results are expressed as single values (*n* = 190), Pearson correlation coefficients, and linear regression lines with 95% CI.

**Figure 2 cells-10-03517-f002:**
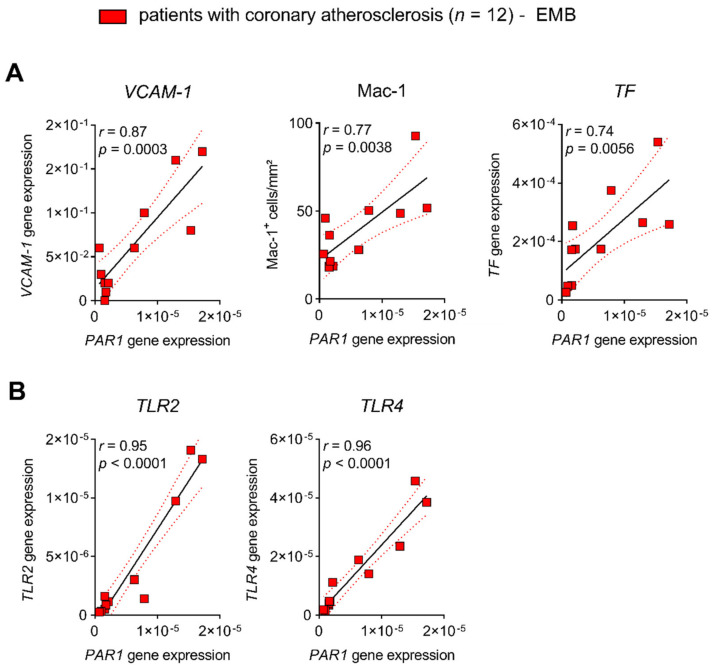
Myocardial PAR1 expression correlates with marker of thrombo-inflammation in patients with coronary atherosclerosis. Relative *PAR1* gene expression in EMBs: (**A**) relative to inflammatory marker *VCAM-1*, *TF* (both relative gene expression), and distribution of CD11b+/Mac-1+ macrophages (IHC); and (**B**) relative to *TLR2* and *TLR4* gene expression. Results are expressed as single values (*n* = 12), Pearson correlation coefficients, and linear regression lines with 95% CI.

**Figure 3 cells-10-03517-f003:**
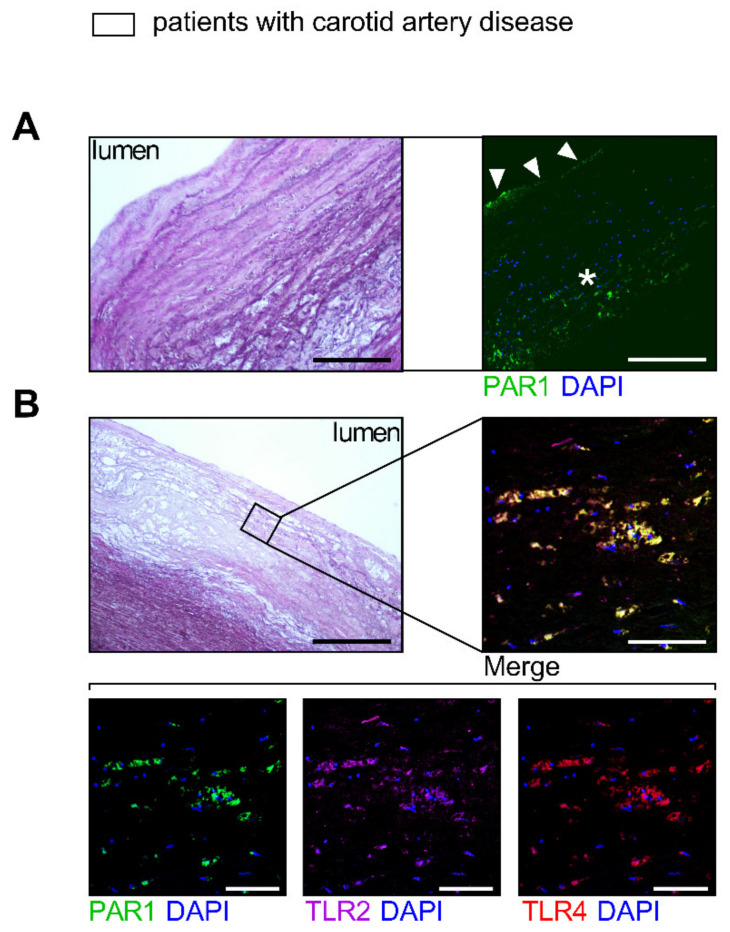
PAR1 is present in human carotid atherosclerotic lesions and colocalizes with TLR2 and TLR4. Representative HE-staining images of human atherosclerotic plaque obtained by CEA (**A**,**B**, left panels). Displayed are the specimen from two patients. Corresponding immunofluorescence staining (**A**,**B**, right panels) reveals (**A**) intraplaque expression of PAR1 (green) within the endothelial layer (arrowheads) and the immune cell-infiltrating area (asterisk). (**B**) Colocalization of PAR1 (green) with TLR2 (magenta) and TLR4 (red). Scale bars (**A**) 160 µm/250 µm and (**B**) 400 µm/100 µm.

**Figure 4 cells-10-03517-f004:**
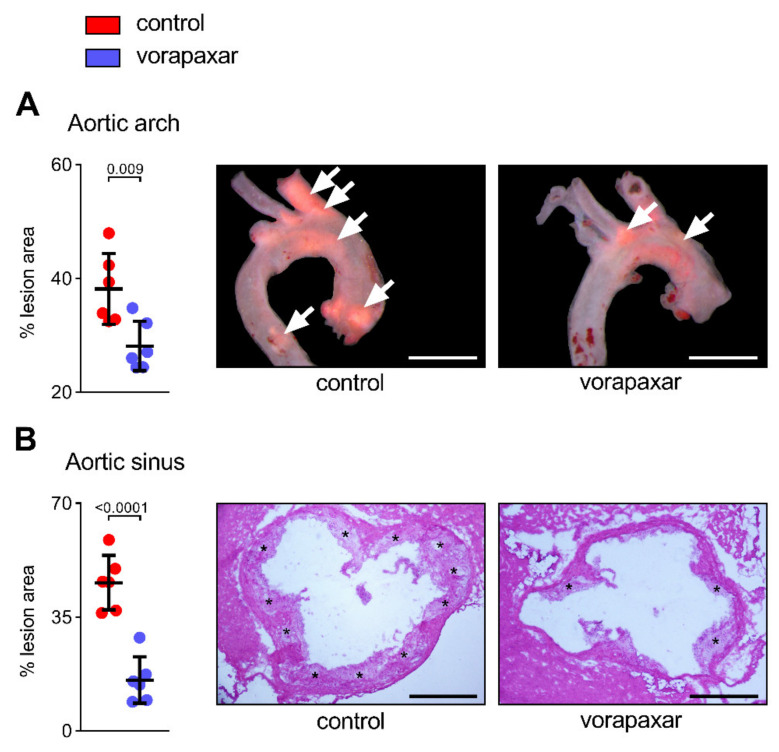
Vorapaxar reduces de novo atherosclerosis in ApoEko mice fed an atherogenic diet. ApoEko mice received a WD (0.21% cholesterol) (control) ± vorapaxar (10 mg/kg WD) for 16 weeks. (**A**) Quantification of plaque areas in aortic arch (arrows), expressed as % Oil red O-positive area. (**B**) Plaque area (%) (asterisks) within aortic sinus (HE-staining images). Representative images, scale bars (**A**) 500 µm and (**B**) 400 µm. Results are expressed as single values (*n* = 6), mean with SD.

**Figure 5 cells-10-03517-f005:**
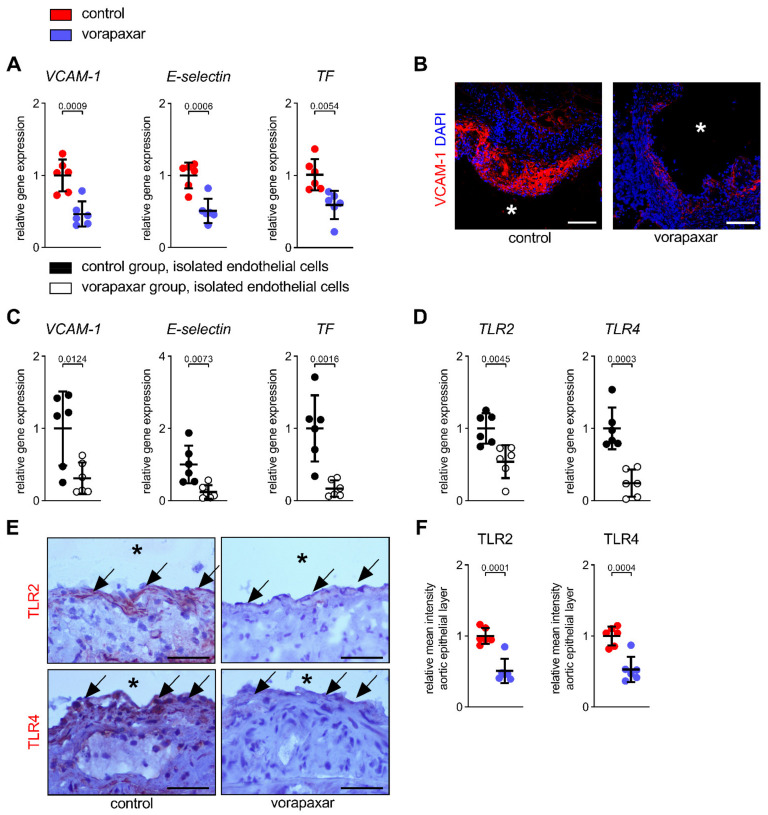
PAR1 inhibition with vorapaxar attenuates vascular inflammation. (**A**) Reduced aortic transcription of *VCAM-1*, *E-selectin*, and *TF*. (**B**) Corresponding immunofluorescence staining of VCAM-1 (red). (**C**,**D**) Endothelial cells were isolated from ApoEko mice on a WD (control) ± vorapaxar (10 mg/kg WD) after 16 weeks. (**C**) Diminished relative gene expression of *VCAM-1*, *E-selectin*, and *TF*. (**D**) Vorapaxar treatment reduced expression of *TLR2* and *TLR4* in isolated endothelial cells and (**E**) within the atherosclerotic plaque covering endothelial layer (arrows) and subepithelial layer (reduced red IHC signal, right panel). (**F**) Quantification of relative mean TLR2 and TLR4 intensity within the atherosclerotic plaque covering the endothelial layer. Representative images, scale bars (**B**) 130 µm and (**E**) 40 µm. Asterisks indicate luminal side. Results are expressed as single values (*n* = 6), mean with SD.

**Figure 6 cells-10-03517-f006:**
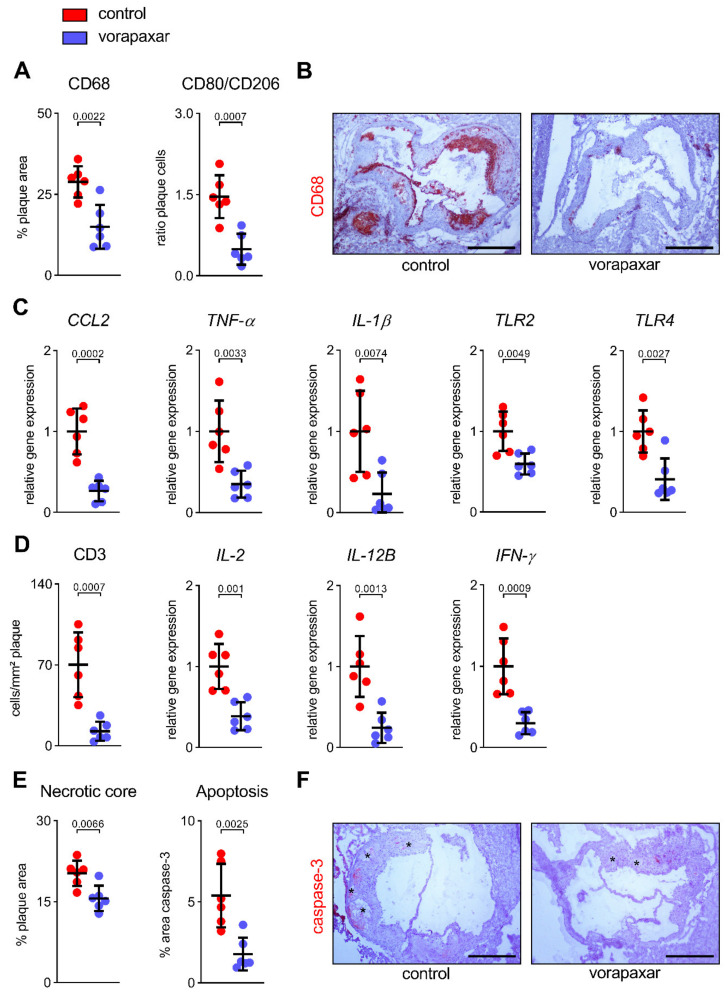
Vorapaxar treatment is associated with an anti-inflammatory cell and cytokine profile within atherosclerotic plaques. (**A**,**B**) Intraplaque distribution of pro-inflammatory CD68-positive cells (% in relation to total plaque area) and the ratio of CD80+/CD206+ cells (M1/M2 macrophages) was reduced under vorapaxar treatment. (**C**) Aortic transcription of chemotactic *CCL2*, proinflammatory cytokines *TNF-α* and *IL-1β*, and innate immune receptors *TLR2* and *TLR4* decreased in the vorapaxar group. (**D**) Presence of CD3+ T cells and aortic gene expression of inflammatory T-cell cytokines (*IL-2*, *IL-12B*, *IFN-γ*) were found to be decreased under the treatment with the PAR1 inhibitor. (**E**,**F**) Reduction of the necrotic core size (% in relation to total plaque area) (asterisk in the corresponding image) and apoptotic plaque area (active caspase-3-positive plaque area) in vorapaxar-treated mice. Representative images, scale bars 400 µm. Results are expressed as single values (*n* = 6), mean with SD.

**Figure 7 cells-10-03517-f007:**
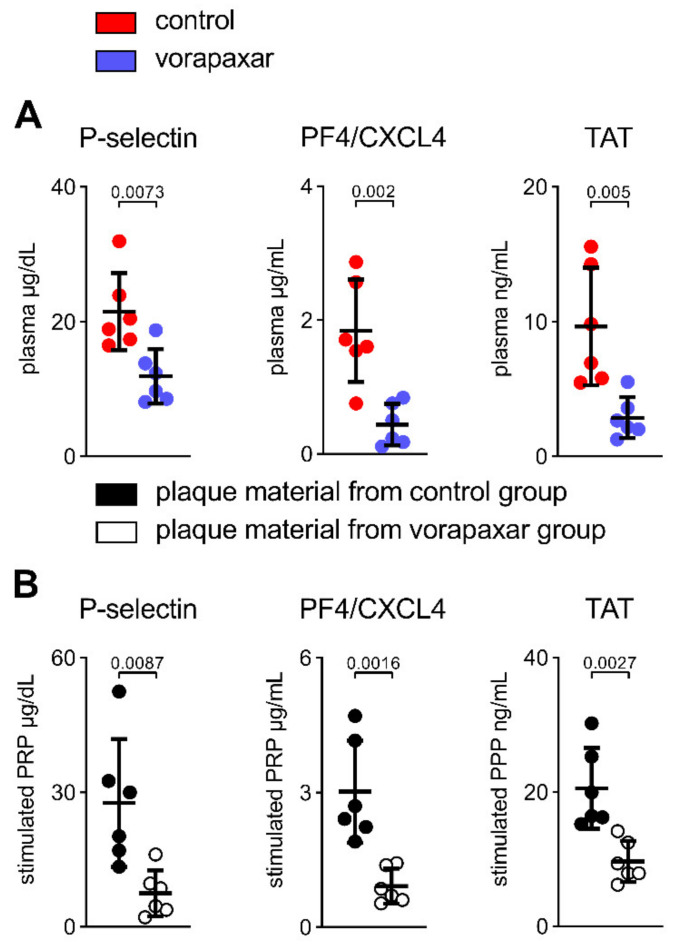
PAR1 inhibition with vorapaxar reduced thrombogenicity in mice. (**A**) In ApoEko mice, addition of vorapaxar reduced circulating markers of platelet activation (P-selectin, PF4/CXCL4) and thrombin activity (TAT). (**B**) Net thrombogenic potential of plaques from either the control or the vorapaxar-treated group. PRP or PPP from B6 wt mice was stimulated with homogenized aortic plaque material. Platelet activation marker (P-selectin, PF4/CXCL4) and thrombin activity (TAT) were measured in supernatant. Results are expressed as single values (*n* = 6), mean with SD.

**Table 1 cells-10-03517-t001:** Baseline characteristics of patients with atherosclerotic disease.

	Biomarker Cohort	EMB Cohort
	(*n* = 190)	(*n* = 12)
Age, yrs	70.3 ± 11.1	62.1 ± 10.5
Male	104/190	9/12
Coronary artery disease	190/190	12/12
History of MI	19/190	1/12
Polyvascular Disease		
Peripheral artery disease	12/190	2/12
Carotid artery disease	51/190	0/12
History of TIA/Stroke	14/190	1/12
Hypertension	123/190	5/12
Diabetes	35/190	3/12
BMI, kg/m²	27.8 ± 5.8	26.6 ± 4.9
CRP, mg/L	7.4 ± 6.7	4.2 ± 4.6

Values are mean ± SD or n. Abbreviations: EMB, endomyocardial biopsy; MI, myocardial infarction.

## Data Availability

The data presented in this study are available on request from the corresponding author. Data from patients are not publicly available due to general data protection regulation.

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
