# Peer review of "Pleiotropic Effects of the Protease-Activated Receptor 1 (PAR1) Inhibitor, Vorapaxar, on Atherosclerosis and Vascular Inflammation"

_cells, 2021, doi:10.3390/cells10123517_

Round 1

Reviewer 1 Report

Very interesting paper about: “Pleiotropic effects of the protease-activated receptor 1 (PAR1) inhibitor, vorapaxar, on atherosclerosis and vascular inflammation”Protease-activated receptor 1 (PAR1) and toll-like receptors (TLRs) are inflammatory mediators contributing to atherogenesis and atherothrombosis. Vorapaxar, which selectively antagonizes PAR1-signaling, is an approved, add-on antiplatelet therapy for secondary prevention. The non-hemostatic, platelet-independent, pleiotropic effects of vorapaxar have not yet been studied
Congratulations for the authors this is a very good paper.

Author Response

Response to Reviewer 1

We thank the reviewer for time and effort assessing the previous version of the manuscript.

Reviewer 2 Report

The authors describe the effect of a PAR1 inhibitor, vorapaxar, which is already in clinical use, on several aspects of atherosclerosis. Although this drug is already in use, the underlying mechanism of its effects are not completely understood. In particular, the authors focused on the formation of atherosclerosis. Therefore they at first analyzed specific marker genes in patients with atherosclerotic disease, followed by a detailed investigation in the ApoEko mouse model treated with vorapaxar. In summary, they observed an increased expression of inflammatory markers as well as genes associated to endothelial activation in these patients. In the ApoEko mice treated with  vorapaxar this increase was suppressed as the authors showed by rt-PCR, immune fluorescence imaging and histology. Further they could observe a reduction of  the formation of foam cells and reduced inflammatory responses in vorapaxar treated mice.  Taken together, they authors investigated in great detail the effects of vorapaxar on atherosclerosis which is important for the field and clinicians.

Some Minor points:

  • The manuscript in general is very detailed described, however a lot of jargon and abbreviations were used which is often not clear to readers outside this field. I don’t know which audience the authors aim for, but a general reader (non-expert in this field) will have a hard time to follow. Maybe some explanations can be added in the result part. I liked the discussion in which the author place the results in context, it might be worth to add some of this in the results. I also often missed a summary sentence after each paragraph/figure/experiment. This would help tremendously.
  • Fig 3: The colocalization of PAR1, TLR2 and 4 is nicely seen here. Can you add a quantification for the colocalization? And an enlarged image would make it easier to evaluate the staining. I appreciate that the authors used human biopsies here, however the same staining was also in Fig. S4 E. So maybe here a detailed quantification of the co-localization can be added?  
  • Fig 4: Please indicate the lesion sites in the images. As far as I understood you used Oli Red O to stain the lesion sites, but then why the treated mice show more intense staining? Do you have a control staining for a wt mouse, maybe this can help to further emphasize and appreciate the lesions. (Maybe also an enlarged image). The same for B – what do I see here? What is the difference in these images. Please indicate with arrows or enlarge the image
  • Fig. 5: Why did you include the staining for VCAM1 but not selectin and TF? Again a quantification of the fluorescence images is missing. Why in 5e you now used histological staining and not immunofluorescence? I am a bit confused here. I also cannot see any difference in TLR2 and 4 localization. (quantification is missing again). Fig. 5C : either you mixed up the colour codes for TF, TLR2 and TLR4 or the text in the results part is not matching.
  • Fig S4: Can vorapaxar rescue/decrease the lipid accumulation in TLR2 and 4 agonist treated cells alone? Or is candesartan needed here? If so why? What happens in candesartan treated cells, is this treatment enough to rescue the effect of thrombin and TLR2 and 4 agonist? If so, this might suggest that PAR1 and TLR2/4 are not in the same pathway , or? Fig. S4E quantification is missing
  • Fig 6A. Why do you use CD68? Please explain in text. Where is the data associated to Cd80/Cd206 graph? What does this mean? Here, in my opinion, more explanation is needed for the reader. Fig. 6F – again very difficult to see/ evaluate.
  • Just out of curiosity : Does vorapaxar change lipid levels in the blood plasma (e.g. LDL, HDL, FFA etc)?

Author Response

Response to Reviewer 2 Comments

We thank the reviewer for time and effort assessing the previous version of the manuscript and providing useful comments to improve its value. We hope that the restructured and rewritten manuscript now clarifies our main conceptual axis. We have addressed all the comments as explained below in the point by point reply.

Point 1: Maybe some explanations can be added in the result part. I liked the discussion in which the author place the results in context, it might be worth to add some of this in the results. I also often missed a summary sentence after each paragraph/figure/experiment.

Response 1: We agree. As per reviewer suggestion, the result section has been rewritten for clarity.

Point 2: The colocalization of PAR1, TLR2 and 4 is nicely seen here. Can you add a quantification for the colocalization?

Response 2: We thank the reviewer for this assistant comment and accordingly added a quantification of the colocalization in the result section.

Point 3: And an enlarged image would make it easier to evaluate the staining.

Response 3: We now provide high-resolution pictures/figures during the submission (zip-file).

Point 4: Fig. S4 E. So maybe here a detailed quantification of the co-localization can be added?

Response 4: We thank the reviewer for this suggestion. Nevertheless, we primarily hypothesize that signaling via PAR1 regulates the expression (and subsequent signaling) of TLR2/4. In our opinion a direct colocalization at the cellular surface might not be a prerequisite for this. Therefore, we would like to suggest not including a colocalization analysis at this stage.

Point 5: Fig 4: Please indicate the lesion sites in the images. As far as I understood you used Oli Red O to stain the lesion sites, but then why the treated mice show more intense staining? Do you have a control staining for a wt mouse, maybe this can help to further emphasize and appreciate the lesions. (Maybe also an enlarged image). The same for B – what do I see here? What is the difference in these images. Please indicate with arrows or enlarge the image.

Response 5: We understand the reviewer’s concern about the Oil red O–staining (Fig. 4A). To highlight the pink glowing lesions (which are completely absent in wildtype mice), we now added arrows. The red spots are remaining blood. To highlight the plaque area in Fig. 4B, we decided to include asterisks. As mentioned above, we now provide high-resolution pictures/figures during the submission (zip-file).

Point 6: Fig. 5: Why did you include the staining for VCAM1 but not selectin and TF?

Response 6: Out intention was to support the data generated from gene expression in aortic specimens. Since our staining protocol worked well with the VCAM-1 antibody (but at this time not with E-selectin or TF due to technical reasons), we decided to include only VCAM-1 staining.

Point 7: Fig. 5: Again, a quantification of the fluorescence images is missing.

Response 7: We agree. We now included a detailed quantification in the results section.

Point 8: Why in 5e you now used histological staining and not immunofluorescence? I am a bit confused here. I also cannot see any difference in TLR2 and 4 localization. (quantification is missing again).

Response 8: The antibodies used in aortic specimen from our mouse model showed no good fluorescence staining quality. Therefore, we decided to perform immunohistochemistry (red staining). In vorapaxar treated mice (right panel), endothelial expression of both TLR2 and TLR4 (red IHC signal) was found to be decreased. In order with your suggestion, we have quantified endothelial TLR2- and TLR4 mean intensity within aortic plaques (new Fig. 5F).

Point 9: Fig. 5C: either you mixed up the color codes for TF, TLR2 and TLR4 or the text in the results part is not matching.

Response 9: Thank you for catching this glaring and confusing error, which we have now corrected in the figure.

Point 10: Fig S4: Can vorapaxar rescue/decrease the lipid accumulation in TLR2 and 4 agonist treated cells alone? Or is candesartan needed here? If so why? What happens in candesartan treated cells, is this treatment enough to rescue the effect of thrombin and TLR2 and 4 agonist? If so, this might suggest that PAR1 and TLR2/4 are not in the same pathway, or?

Response 10: We thank the reviewer in supporting our intention. In order with your suggestion, we included the data of lipid accumulation in THP-1 treated with thrombin+TLR2/4 agonists + candesartan (but without vorapaxar) and thrombin+TLR2/4 agonists + vorapaxar (but without candesartan). In contrast to vorapaxar, candesartan (that decreases TLR2/4-expression), diminished lipid accumulation but was not able to rescue the thrombin-initiated effect. We have added this information to Supplemental Figure 4B.

Point 11: Fig. S4E quantification is missing.

Response 11: We agree. We now included a detailed quantification (now Suppl. Fig. 4F).

Point 12: Fig 6A. Why do you use CD68? Please explain in text. Where is the data associated to Cd80/Cd206 graph? What does this mean? Here, in my opinion, more explanation is needed for the reader.

Response 12: We thank the reviewer for this suggestion and have revised this part for clarity. CD68 may be used as a pan-macrophage or M1 marker. Since CD68 can be also expressed by a variety of pro-inflammatory plaque residing cells (SMCs etc.), CD68 might be seen as a global marker of inflammatory burden within atherosclerotic plaques. CD68 is omnipresent in atherosclerotic plaques. Therefore, CD68 can only be quantified in relation to total plaque area. In order to have a more specific approach we used CD80 as a marker for M1 pro-inflammatory macrophages and CD206 as a marker for M2 anti-inflammatory macrophages.

Point 13: Fig. 6F – again very difficult to see/ evaluate.

Response 13: As mentioned above, all pictures are now provided with the highest possible resolution.

Point 14: Just out of curiosity: Does vorapaxar change lipid levels in the blood plasma (e.g. LDL, HDL, FFA etc)?

Response 14: The reviewer raises an interesting and important question. As shown in Supplemental Figure 1, vorapaxar does not alter food intake, weight gain, or circulating cholesterol. PAR2 has been linked to obesity and diabetes. Furthermore, PAR1 and hematopoietic cell TF are required for liver inflammation and steatosis in mice fed a western diet. Therefore, it is likely that vorapaxar might have some impact on the lipoprotein profile. We are addressing this question in a recent project, which is still ongoing.
